# MPformer: Advancing Graph Modeling Through Heterophily Relationship-Based Position Encoding

## Abstract

Graph transformer model integrates the relative positional relationships among nodes into the transformer architecture, holding significant promise for modeling graph-structured data. They address certain limitations of graph neural networks (GNNs) in leveraging information from distant nodes. However, these models overlooked the representations of neighboring nodes with dissimilar labels, i.e., heterophilous relationships. This limitation inhibits the scalability of these methods from handling a wide range of real-world heterophilous datasets. To mitigate this limitation, we introduce MPformer, comprising the information aggregation module called Tree2Token and the position encoding module, HeterPos. Tree2Token aggregates node and its neighbor information at various hop distances, treating each node and its neighbor data as token vectors, and serializing these token sequences. Furthermore, for each newly generated sequence, we introduce a novel position encoding technique called HeterPos. HeterPos employs the shortest path distance between nodes and their neighbors to define their relative positional relationships. Simultaneously, it captures feature distinctions between neighboring nodes and ego-nodes, facilitating the incorporation of heterophilous relationships into the Transformer architecture. We validate the efficacy of our approach through both theoretical analysis and practical experiments. Extensive experiments on various datasets demonstrate that our approach surpasses existing graph transformer models and traditional graph neural network (GNN) models.

## 1 Introduction

While graph neural networks exhibit proficiency in handling non-Euclidean data, they encounter challenges related to limited receptive fields and the effective utilization of distant node information. Conversely, the Transformer architecture has gained widespread adoption in natural language processing and computer vision tasks due to its extensive modeling capabilities across long ranges. To address the challenges faced by graph neural networks, researchers initiated investigations into extending the Transformer architecture for graph data. This approach aimed to inherit GNN's competence in processing non-Euclidean data while leveraging the Transformer architecture's expansive receptive field to compensate for GNN's limitations in utilizing distant node information. However, when dealing with graph-structured data, there are complex attributes involved, such as structural topology and attribute features, that cannot be directly encoded as tokens in the Transformer architecture. As a result, a significant amount of research in this area focuses on integrating graph structure information into the Transformer architecture. Graph-structured data frequently includes complex attributes like structural topology and attribute features, which cannot be directly encoded within the Transformer architecture. Consequently, extensive research in this phase has concentrated on incorporating graph structural information into Transformer architectures. Some of these studies concentrate on integrating graph neural networks with the Transformer architecture. For instance, GraphTrans(Wu et al. (2021)) and GraphiT(Mialon et al. (2021)) employ GNNs to create structure-aware representations from original features. They subsequently leverage the computational aspects of the Transformer architecture to handle all pairwise node interactions, thereby enhancing the model's global inference capability. But these approaches do not accurately model sequential information jumps, which may result in the blending of features with varying orders. Furthermore,

methods that integrate graph neural networks into the Transformer architecture demand thorough parameter tuning, adding complexity to the model training process.

Consequently, some researchers have investigated techniques for directly incorporating graph topology information into the transformer architecture. Dwivedi & Bresson (2020) utilizes the $k$ smallest non-zero eigenvalue of the Laplace matrix with its corresponding eigenvector for positional embedding. similarly, Hussain et al. (2021) employs singular value decomposition on the adjacency matrix, utilizing the left and right singular vectors associated with the top $r$ singular values as a means of representing position encoding. Furthermore, in contrast to the compression of graph structural data into position encoding via Laplacian matrices, Graphormer(Ying et al. (2021)) employs degree centrality derived from the adjacency matrix as its positional encoding method. Additionally, Graph-BERT (Zhang et al. (2020)) introduces three distinct types of position encoding to integrate node position information: absolute position encoding based on the Weisfeiler-Lehman algorithm, absolute position encoding, and relative position encoding based on intimacy and shortest path distance. In contrast to embedding graph structure information into positional encoding, alternative approaches leverage graph structure information as a bias within the attention matrix. For instance, Ying et al. (2021), Khoo et al. (2020), and Zhao et al. (2021) incorporate pairwise information as a bias when computing the attention matrix. Nevertheless, these approaches still overlook the possible heterogeneous relationships among interconnected nodes, leading to models struggle to handle heterophily.

To enhance modeling capabilities for heterophilous data, we present MPformer. Differing from prior graph transformers that treat nodes as independent tokens, MPformer leverages both individual node information and that of its neighbors to form a new sequence. This is achieved through the Tree2Token module, which regards each node and its neighbors as tokens, thus creating a new sequence that preserves neighborhood information to facilitate transformer module training. Furthermore, to incorporate graph topological information and heterophilous relationships into the model, we introduce the HeterPos position encoding module for MPformer. HeterPos uses the distance of the shortest path between a node and its neighboring nodes to determine their positional relationship. It takes into account the differences in features between nodes and their neighbors to incorporate heterophilous relationships into the model.

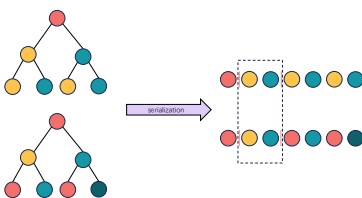

Figure 1: a serializing case show two cases under heterophily.

To facilitate the comparison of our approach with existing methods, we provide an illustrative example of a heterogeneous graph, as depicted in Fig.1. Based on the figure, accurately classifying the ego node (root node) within the graph can be challenging when relying solely on the inter-node degree matrix and relative positional relationships. However, our approach enables the model to identify similarities among node neighbors by treating a node and its neighbors as tokens and serializing them. This enhances the model's capacity to capture hop-wise semantic correlations, ultimately improving its ability to model heterophilous graph. Our model yields the following practical contributions:

- Our model is improved by creating new tokens based on nodes and their neighbors, gathering information from nearby nodes at varying hop distances in a flexible way. This approach ensures versatility by avoiding the stacking of node information from different hops and provides a more detailed understanding of the relationships between nodes and their neighbors. By integrating localized structural information into the model, our methodology results in better performance on heterophilous graphs.

- We develop an innovative position encodings approach that leverages hop counts, node representation, and neighboring node representation to seamlessly incorporate shortest path distances and heterophilous relationships into the Transformer architecture, offering fresh insights for extending the graph transformer model to heterogeneous graphs in the future.

- We validate our approach using multiple datasets and consistently achieved superior performance compared to the current state-of-the-art method. These results demonstrate the high effectiveness and reliability of our approach in achieving superior results.

## 2 MPFORMER

In this section, we introduce MPformer, illustrated in Figure 2. To enhance its effectiveness in handling graph-structured data, we begin by introducing the Tree2Token neighborhood aggregation module, which constructs new sequences for nodes and their neighbors. Next, we present HeterPos, a novel positional encoding method designed to capture relative positional and heterophily relationships within this sequence and integrate them into the Transformer architecture. We also conduct a theoretical analysis of HeterPos to highlight its exceptional properties. Finally, we present the algorithmic flow of MPformer, providing a detailed explanation of each step in the process.

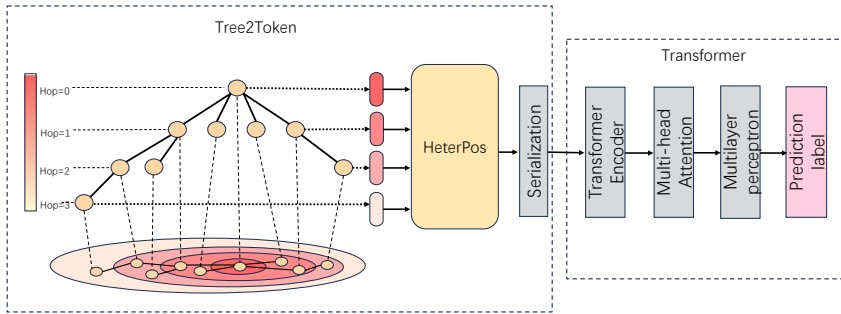

Figure 2: This figure depicts the architecture of the MPformer model.

### 2.1 TREE2TOKEN

Utilizing neighbor information from different hops has proven to be highly effective in capturing hop-wise semantic relations, significantly enhancing the model's ability to manage graph-structured data(Abu-El-Haija et al. (2019), Zhu et al. (2020), Song et al. (2023)). To leverage this exceptional capability, we introduce Tree2Token, a mechanism that considers neighbor information from various hops.

Consider a node $v$, and we view the node $v$ and its neighbor nodes at each hop as a rooted-tree hierarchy as illustrated in Fig.2. $\mathcal{H}^k(v)$ represents the collection of nodes on a shortest path distance of $k$ from node $v$. All nodes within this set constitute the same $k$-th hop. Specifically, $\mathcal{H}^0(v)$ is the set that includes only node $v$ itself, and it's referred to as the 0-th hop. Initially, we aggregate information from neighboring nodes that reside at the same hop:

$$\mathbf{x}_v^k = \phi(\mathcal{H}^k(v)) \tag{1}$$

Here, $\mathbf{x}_v^k \in \mathbb{R}^d$ denotes the node representation at the $k$-th hop, and $\phi$ represents the aggregation function. This aggregation operation helps prevent the overlapping of neighbor information from different hops in the previous message passing. Referring to equation 1, we generate token vectors for each hop of node $v$ and create a sequence, $\mathcal{S}_v^k = (\mathbf{x}_v^0, ..., \mathbf{x}_v^k)$, representing the neighborhood information of node $v$, with $k$ as a hyperparameter. For all nodes on the graph $\mathcal{G}$, we can construct a matrix $\mathbf{X}_{\mathcal{G}}^k = (\mathcal{S}_1^k, ..., \mathcal{S}_n^k)^\top \in \mathbb{R}^{n \times (k+1)d}$, with $n$ representing the number of nodes.

To provide a more detailed explanation of our computational procedure, we begin by introducing two operators: one for regularization denoted as $\mathbb{N}$, and the other for binarization denoted as $\mathbb{B}$.

$$\mathbb{N}(\mathbf{A}) = \mathbf{D}^{-\frac{1}{2}} \mathbf{A} \mathbf{D}^{-\frac{1}{2}},$$

$$\mathbb{I}(\mathbf{A}) = \begin{cases} 1 & \text{if } \mathbf{A}_{ij} \geq 0 \\ 0 & \text{otherwise} \end{cases} \tag{2}$$

In this context, $\mathbf{D}$ represents a diagonal matrix, and each diagonal element $d_{ii}$ in $\mathbf{D}$ corresponds to the sum of the elements in the $i$th row of matrix $\mathbf{A}$. Specifically, when $\mathbf{A}$ serves as an adjacency matrix, $\mathbf{D}$ takes on the role of a degree matrix. Consequently, we can derive

$$\mathbf{A}_{norm}^{(k)} = \mathbb{N}(\mathbb{B}(\mathbf{A}^k - \mathbf{A}^{k-1} - \mathbf{I}))$$
$$\mathbf{X}_{\mathcal{G}}^k = [\mathbf{X}_{\mathcal{G}}^0, \mathbf{A}_{norm}^{(1)} \mathbf{X}_{\mathcal{G}}^0, ..., \mathbf{A}_{norm}^{(k)} \mathbf{X}_{\mathcal{G}}^0] \tag{3}$$

where $\mathbf{I} \in R^{n \times n}$ is identity matrix. In practice, for more efficient aggregation of level information, we employ an aggregation method similar to Zhu et al. (2020):

$$[\mathbf{A}^{(1)}_{norm}\mathbf{X}^k_{\mathcal{G}}, \mathbf{A}^{(2)}_{norm}\mathbf{X}^k_{\mathcal{G}}] \tag{4}$$

The Tree2Token approach provides two significant advantages. Firstly, this aggregation method encodes information about each level of a node's neighbors as representations, facilitating the capture of semantic correlations between these levels—typically overlooked in conventional GNNs. Additionally, this aggregation method mitigates the issue of stacking between different hop nodes , a challenge in traditional message passing methods.

## 2.2 HETERPOS

Initially, positional encoding techniques were used in sequence data to provide models with positional information about tokens within input sequences. This improved the models' ability to understand and process such data. However, graph-structured data is more complex than sequential data. Unlike sequence data, where each token has a clear direction, nodes in graph-structured data are dispersed in a complex spatial arrangement, making it difficult to establish clear directions. As a result, the conventional positional encoding approach is not suitable for graph-structured data.

Recently, several positional encoding approaches have emerged for graph-structured data, with the aim of incorporating graph-structural information into the Transformer architecture to enhance the model's capacity for graph data modeling. However, most of the current positional encoding methods designed for graph Transformer models focus only on the relative positional relationships between nodes and basic topological attributes, such as node degrees. They tend to overlook situations where nodes have different labels compared to their connected nodes. This oversight can lead to problems related to heterophily.

To deal with heterophily, the key is to focus on the presence of distinct labels among connected nodes. When designing position encoding, our goal is to capture the variations in features between a node and its neighboring nodes. Additionally, we must consider the diminishing influence of neighboring nodes on the ego node as the distance between them increases. To achieve this goal, we propose the HeterPos module. For each node $v$, its feature vector, represented as $\mathbf{x}^k_v \in \mathbb{R}^d$, with $k$th hop neighbors, is encoded as follows:

$$\mathrm{PE}(\mathbf{x}^k_v) = [\mathbf{x}^k_v, \sin(c\mathbf{x}^k_v e^{-k \ln(10000/d)}), \cos(c\mathbf{x}^k_v e^{-k \ln(10000/d)})]\mathbf{w}_{\mathrm{PE}} \tag{5}$$

where $k$, and $d$ are hyperparameters, $c$ is a constant, $\mathbf{w}_{\mathrm{PE}} \in \mathbb{R}^{3d}$.

### 2.2.1 WHAT HETERPOS CAN TELL US ABOUT POSITION?

To provide a more comprehensive understanding of our proposed position encoding, we utilize the analytical approach of position encoding as described in (Shaw et al. (2018), Dai et al. (2019)).These encoded node features are integrated into the section of the Transformer architecture that calculates the self-attention matrix, leading to the formulation of equation 6. For a more detailed explanation and step-by-step procedure, please consult the Appendix A.1.

$$\begin{aligned} &[\mathbf{x}^k_v, \sin(c\mathbf{x}^k_v e^{-k\ln(10000/d)}), \cos(c\mathbf{x}^k_v e^{-k\ln(10000/d)})]^\top \cdot \\ &[\mathbf{x}^j_v, \sin(c\mathbf{x}^j_v e^{-k\ln(10000/d)}), \cos(c\mathbf{x}^j_v e^{-j\ln(10000/d)})]. \end{aligned} \tag{6}$$

It is important to mention that one of the factors involved in the equation expansion is:

$$\begin{aligned} &\sin(c\mathbf{x}^k_v e^{-k\ln(10000/d)})\sin(c\mathbf{x}^j_v e^{-\alpha k\ln(10000/d)})+ \\ &\cos(c\mathbf{x}^k_v e^{-j\ln(10000/d)})\cos(c\mathbf{x}^j_v e^{-\alpha j\ln(10000/d)}). \end{aligned} \tag{7}$$

Thus by trigonometric properties we have:

$$\cos(c\mathbf{x}^k_v e^{-k\ln(10000/d)} - c\mathbf{x}^j_v e^{-j\ln(10000/d)}). \tag{8}$$

From the analysis above, our positional coding provides two clear benefits: (1) It accurately captures the relative differences between the central node and its $k$-th hop neighbors, allowing for precise adjustments of these differences through the parameters $d$. (2) As the parameter $k$ increases, the influence of the $k$-th hop neighboring nodes on the central node decreases, enabling the model adapt to the relative positional changes between a node and its neighbors. The results in Fig. 6 validates these viewpoints.

### 2.3 A RESCALED REGULARIZATION OF $\mathbf{W}_{\text{PE}}$ FOR BETTER GENERALIZATION

In order to enhance the generalization ability of the HeterPos module and at the same time give it more flexibility, we further analyze and discuss HetePos from a theoretical perspective in this section.

**Definition 1** *For a given class $\mathcal{F}$ of vector-valued functions, the covering number $\mathcal{N}_\infty(\mathcal{F}; \epsilon; \{z^{(i)}\}^{m}_{i=1}; \|\cdot\|)$ is defined as the smallest collection such that for any $f \in \mathcal{F}$, there exists $\hat{f} \in \mathcal{C}$ satisfying:*

$$\max_i \|f(z^{(i)}) - \hat{f}(z^{(i)})\| \le \epsilon \tag{9}$$

*Furthermore, it is denoted as:*

$$\mathcal{N}_\infty(\mathcal{F}, \epsilon, m, \|\cdot\|) = \sup_{z^{(1)},...,z^{(m)}} \mathcal{N}_\infty(\mathcal{F}; \epsilon; z^{(1)}, ..., z^{(m)}, \|\cdot\|). \tag{10}$$

Based on the Definition 1, we introduce a lemma about the Capacity of a Transformer head without Position encoding:

**Lemma 1 (Edelman et al. (2022))** *For all $\epsilon > 0$ and $\mathbf{x}^{(1)}, ..., \mathbf{x}^n \in \mathbb{R}^d$ such that $\|\mathbf{x}^{(i)}\| \le B_x$ for all $i \in [n]$. Then the covering number of $\mathcal{F}$ satisfies*

$$\log \mathcal{N}_\infty(\mathcal{F}; \epsilon; \mathbf{x}^{(1)}, ..., \mathbf{x}^{(n)}, \|\cdot\|) \lesssim (L_\sigma B_x)^2 \cdot \log(nd) \cdot \frac{((B_V)^{\frac{2}{3}} + (B_{QK} B_V B_x)^{\frac{2}{3}})^3}{\epsilon^2}. \tag{11}$$

Here, $B_V$ is the upper bound of the weight matrix $\|\mathbf{W}_V\|$ for computing values in the attention mechanism, and $B_{QK}$ is the upper bound of the weight matrix $\|\mathbf{W}_Q \mathbf{W}_K\|$ for computing queries and keys in the attention mechanism. Based on the above Lemma 1, we now consider the inclusion of HeterPos for Transformer model, then Theorem 1 is given in this case.

**Theorem 1** *Assume that the activation function $\sigma$ is L-Lipschitz, and for all $\mathbf{x}^{(1)}, ..., \mathbf{x}^{(n)} \in \mathbb{R}^d$, there exists $\|\mathbf{X}^{(i)\top}\|$ for all $i \in [n]$. Then, the covering number of $\mathcal{F}$ satisfies:*

$$\log \mathcal{N}_\infty(\mathcal{F}(\mathbf{X}; \epsilon; \mathbf{x}^{(1)}, ..., \mathbf{x}^{(n)}, \|\cdot\|)) \lesssim \frac{((B_V)^{2/3} + (2B_{Q,K} B_W B_V (B_X + 1))^{2/3})^3}{\epsilon^2} \cdot (L_\sigma(B_X + 1))^2 \log(nd) \tag{12}$$

*Further, when the loss function is b-bounded loss function that is L-Lipschitz in its first argument, and for all $\mathbf{x}^{(i)}$ obeys the distribution $D$, we can get the generalization error bound satisfies*

$$|\text{risk}(f; D) - \hat{\text{risk}}(f; (\mathbf{x}^{(i)}, y^{(i)})^m_{i=1})| \le 4cL\sqrt{\frac{N_\mathcal{F}}{n}}(1 + \log(A\sqrt{\frac{n}{N_\mathcal{F}}})) + 2b\sqrt{\frac{\log(1/\delta)}{2n}} \tag{13}$$

*where $N_\mathcal{F} = ((B_V)^{2/3} + (2B_{PE} B_{Q,K} B_W B_V (B_X + 1))^{2/3})^3 \cdot (L_\sigma B_{PE}(B_X + 1))^2 \log(nd)$, $|f| \le A$ for all $f \in \mathcal{F}$ and $c > 0$ is a constant, $B_W$ is the upper bound of the weight matrix $\|\mathbf{W}_{PE}\|$.*

**Remark** Theorem 1 provides insights into an underlying trade-off, specifically, when the upper bound 12 becomes larger it may leads to a larger covering number, which implies a larger hypothesis space, and thus may lead to a smaller empirical risk, i.e., exhibit a smaller training loss. Meanwhile, according to Eq.13, a larger covering number implies a larger generalized Gap (L.H.S of Eq.13), and this means that the model's performance on the training and test sets may be more inconsistent, i.e., more unstable. Therefore, in order to further improve the generalization performance of the model, we need to find a trade-off between stability and empirical risk minimization to ensure the generalization ability of this model. Note that $B_W$ in Eq.12 is directly related with HeterPos.

Based on the analysis above, we propose a rescaled regularization for HeterPos to enhance its flexibility of trade-off the stability and generalization ability. The loss function is shown as the equation below:

$$\mathcal{L} = NLL + \alpha\|\mathbf{W}_{\text{PE}}\|, \tag{14}$$

where $\alpha$ is a optional scaling factor to adjust the restrict on the value of $B_W$. To validate our theoretical analysis, we conduct experiments on two datasets, randomly selected in heterophilous and homophilous datasets, respectively. We plotted the curves representing empirical risk and generalized risk while varying the value of $\alpha$.

The experimental results are presented in the Fig.3, revealing a clear trade-off between generalization risk and empirical risk, which matched with our analysis. Additionally, we can achieve a balance between stability and generalization ability by tuning the scaling factor $\alpha$. Specifically, the best choice of $\alpha$ is $1e - 4$ as shown in the figure.

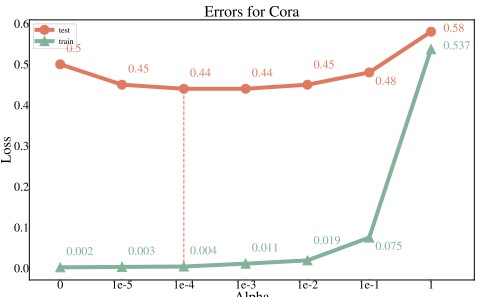 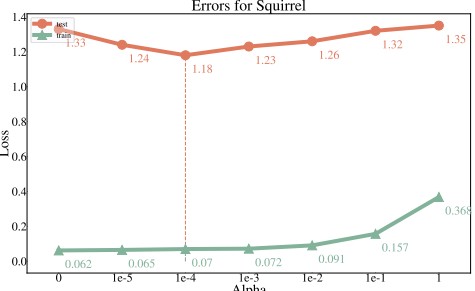

Figure 3: The figure shows the training and testing errors for different limits on the upper bound of $\mathbf{W}_{\mathrm{PE}}$ for the model cora and under the Squirrel dataset.

## 3 EXPERIMENTS

### 3.1 EXPERIMENTAL SETUP

**Datasets & Setup** We assess the performance of our model on both homophilic and heterophilic datasets. Specifically, we utilize two homophilic citation network datasets, namely Cora and Citeseer(Sen et al. (2008)), as well as six heterophilic network datasets: Cornell, Texas, Wisconsin, Actor, Chameleon, and Squirrel (Pei et al. (2020); Rozemberczki et al. (2021)). In the context of citation networks, nodes correspond to documents, and edges represent citations between them. Node features are represented as bag-of-words representations of the documents. In contrast, in web networks, nodes represent web pages, and edges symbolize hyperlinks between them. Node features are also represented as bag-of-words representations of the web pages. To quantify the level of homogeneity for each dataset, we calculate the marginal homophily score $\mathcal{H}(G)$ (Yan et al. (2022)). The results of these calculations are summarized in Table 1. Our optimization strategy employs the Adam optimizer, with L2 regularization applied to $\mathbf{W}_{\mathrm{PE}}$. Across each dataset, we employ a two-headed attention mechanism and a single layer of the Transformer architecture. We perform a comprehensive hyperparameter tuning process for all models using grid search. Detailed information can be found in the appendix.

Table 1: Homophily, number of nodes, number of edges, and number of node classes for each dataset item

|  | Cora | Citeseer | Cornell | Texas | Wisconsin | Actor | Chameleon | Squirrel |
|---|---|---|---|---|---|---|---|---|
| $\mathcal{H}(G)$ | 0.81 | 0.74 | 0.3 | 0.11 | 0.21 | 0.22 | 0.23 | 0.22 |
| Nodes | 2,708 | 3,327 | 183 | 183 | 251 | 7,600 | 2,277 | 5,201 |
| Edges | 5,278 | 4,676 | 280 | 295 | 466 | 26,752 | 31,421 | 198,493 |
| Classes | 6 | 7 | 5 | 5 | 5 | 5 | 5 | 5 |

### 3.2 HOMOPHILY AND HETEROPHILY

We evaluated the performance of our model on eight datasets using the same data splitting approach outlined by Yan et al. (2022). The results are presented in Table 2, which displays the mean accuracy and standard deviation across ten random data partitions for the test set. Our

baseline models include traditional Graph Neural Network (GNN) architectures, such as vanilla Graph Convolutional Networks (GCN) (Kipf & Welling (2016)), Graph Attention Networks (GAT) (Veličković et al. (2017)), and GraphSAGE (Hamilton et al. (2017)). Additionally, we incorporated models specifically designed for heterophily, such as Heterogeneous Heterophily Graph Convolutional Networks (H2GCN) (Zhu et al. (2020)), Graph Partition Regularized Graph Neural Networks (GPRGNN) (Chien et al. (2020)), and Generalized Graph Convolutional Networks (GGCN) (Yan et al. (2022)). We also compared two methods for aggregating information from different hop nodes: MixHop(Abu-El-Haija et al. (2019)) and JK-net(Xu et al. (2018)). Furthermore, we evaluated two state-of-the-art GNN methods, namely Graph Convolutional Networks with Information Integration (GCNII) (Chen et al. (2020)) and Ordered Graph Neural Networks (OrderedGNN) (Song et al. (2023)). To further explore the capabilities of our model, we also tested three graph Transformer models: Spectral Attention Network (SAN) (Kreuzer et al. (2021)), fast Graph Transformer Network (FGTN) (Yun et al. (2022)), and Specformer (Bo et al. (2023)).

Table 2: Results on real-world node classification tasks.

|  | Cora | Citeseer | Cornell | Texas | Wisconsin | Actor | Chameleon | Squirrel |
|---|---|---|---|---|---|---|---|---|
| GCN | 87.1±1.01 | 76.6±0.67 | 60.54±5.30 | 55.14±1.10 | 51.96±3.06 | 27.34±1.10 | 64.5±2.24 | 53.05±2.01 |
| GAT | 87.69±1.23 | 76.55±1.10 | 61.08±5.05 | 53.51±6.63 | 49.8±4.09 | 27.63±0.89 | 59.85±2.50 | 40.64±1.55 |
| GraphSAGE | 86.90±1.04 | 76.04±1.30 | 75.95±5.01 | 82.43±6.14 | 81.18±5.56 | 34.23±0.99 | 58.73±1.68 | 41.61±0.74 |
| H2GCN | 86.84±1.20 | 75.91±1.56 | 74.32±5.28 | 80.27±7.23 | 80.20±4.98 | 32.13±1.00 | 62.08±2.15 | 31.58±1.86 |
| GPRGCN | 87.59±1.18 | 76.98±1.67 | 77.84±8.11 | 74.86±4.36 | 82.35±4.21 | 33.94±1.22 | 45.79±1.71 | 28.63±1.24 |
| GGCN | 87.55±1.05 | 76.78±1.45 | 84.05±6.63 | 82.7±4.55 | 86.67±3.29 | 37.53±1.56 | 71.16±1.84 | 55.12±1.58 |
| OrderedGNN | 85.31±0.75 | 73.81±1.73 | 74.05±4.73 | 86.21±4.12 | 85.29±3.36 | 37.5±1.00 | 71.29±2.29 | 59.8±1.96 |
| MixHop | 87.61±0.85 | 76.26±1.33 | 73.51±6.34 | 77.84±7.73 | 75.88±4.90 | 32.22±2.34 | 60.50±2.53 | 43.80±1.48 |
| JK-Net | 85.96±0.83 | 76.05±1.37 | 75.68±4.03 | 83.78±2.21 | 82.55±4.57 | 35.14±1.37 | 63.79±2.27 | 45.03±1.73 |
| GCNII | 88.51±1.25 | 77.1±1.48 | 74.32±3.79 | 71.08±3.38 | 72.75±3.40 | 37.46±1.30 | 61.86±3.04 | 37.03±1.58 |
| Geom-GCN | 85.35±1.57 | 78.02±1.15 | 60.54±3.67 | 66.76±2.72 | 64.51±3.66 | 31.59±1.15 | 60.00±2.81 | 38.15±0.92 |
| Specformer | 87.40±1.01 | 74.92±0.94 | 72.16±3.32 | 79.19±4.34 | 81.37±2.33 | 27.01±1.02 | 70.13±2.33 | 58.27±2.12 |
| FGTN | 86.52±1.32 | 80.66±2.21 | 83.78±6.42 | 81.08±5.27 | 86.27±4.15 | 37.30±0.67 | 50.00±2.91 | 31.12±1.78 |
| OURS | 88.13±1.12 | 77.28±1.57 | 86.76±5.68 | 90±6.16 | 90.2±9.49 | 37.55±1.44 | 75.07±1.96 | 66.5±4.11 |

The proposed MPformer model has outperformed all existing methods, achieving state-of-the-art (SOTA) results on both homophilic and heterophilic datasets. On homophilic datasets, our approach is comparable to the FGTN model. However, the FGTN model exhibits inferior performance on heterophilic datasets. Notably, on the Squirrel dataset, our method demonstrates a performance superiority of over 35% compared to the FGTN model. We have observed that MPformer's enhancement on heterophilic datasets is more pronounced than on homophilic graphs. This phenomenon may be attributed to the fact that in homophilic graphs, the distinctions between nodes and their neighboring nodes are comparatively smaller, thereby constraining the advantage of MPformer in leveraging node and neighbor node disparities. Furthermore, our investigation revealed that contemporary graph Transformer models do not exhibit significant improvements over Graph Neural Network (GNN) models in the context of node classification tasks, potentially due to inadequate attention to local information.

### 3.3 EFFECTIVENESS STUDY OF HETERPOS

In order to analyze the effectiveness of our proposed position encoding, we performed a series of ablation studies on all datasets.

**Setups**  To validate the effectiveness of HeterPos, we employed the Tree2Token module for information aggregation across all eight datasets. We conducted a performance comparison between models utilizing HeterPos position encoding and those employing various other position encoding methods, including absolute position encoding (Vaswani et al. (2017)), relative position encoding (Dai et al. (2019)), learnable position encoding (Devlin et al. (2019)), rotational position encoding(Su et al. (2021)), and Laplacian position encoding(Dwivedi & Bresson (2020)). Each of these position encoding techniques was applied to the sequences generated by the Tree2Token module.

Notably, Tree2Token treats a node and its neighboring features as tokens, constructing a new sequence based on them. For both absolute and rotational position encoding, we assigned hop numbers as token position numbers. In the case of relative position encoding, we quantified the relative position relationships between nodes using the distances between a node and its neighboring nodes. It is important to emphasize that to ensure a fair and unbiased comparison, we maintained uniform parameter configurations when testing all position encoding methods.

**Analysis** Figure 4 presents our experimental findings. The results vividly demonstrate the considerable superiority of HeterPos over alternative position encoding methods. Furthermore, both relative position encoding and learnable position encoding exhibit notably superior performance compared to other encoding techniques. This phenomenon can be attributed to the unique characteristics of these encoding methods. Relative position encoding quantifies the relative positional relationships between nodes by considering the distances between a node and its neighboring nodes. On the other hand, learnable position encoding offers enhanced flexibility in comparison to alternative methods. This flexibility enables the model to capture valuable structural information from the graph during training, thereby enhancing its overall performance. In contrast, rotational position encoding demonstrates inferior effectiveness compared to position encoding-free methods. This outcome stems from the inherent complexity of graph-structured data, where node distribution in space lacks a clear, ordered pattern, making it challenging to establish a definite directional relationship between nodes and their neighbors. Encoding based solely on hop numbers proves inadequate in defining such directionality, leading to the inclusion of extraneous and detrimental information within the model, thereby compromising its performance.

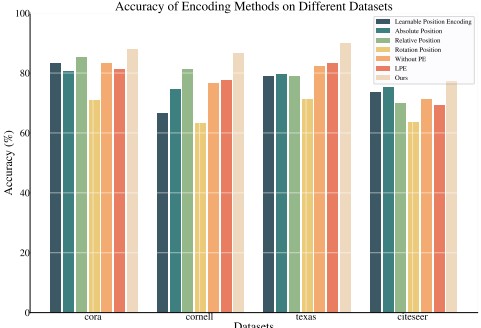 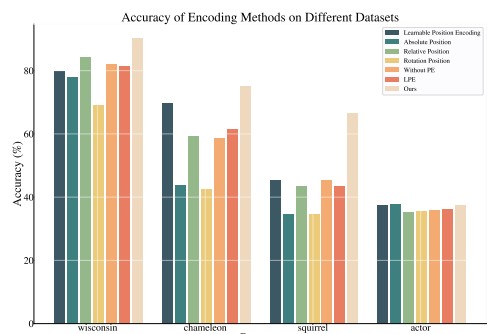

Figure 4: Performance of different location coding methods on each dataset

### 3.3.1 PARAMETER STUDY

In order to fully evaluate the impact of our proposed position encoding parameters on the model performance, we experimentally illustrate each parameter in HeterPos separately.

**Constant $c$** To investigate the influence of different values of the constant $c$ on HeterPos, we set the number of hops in HeterPos to 1 and dimensions $k$ and $d$ to 128, respectively. We sequentially vary the constant $c$ within the range of 0 to 100 and assess how Equation 8 changes in response to $c$ variations. The results are depicted in Figure 5 and Figure 6. From figures, we observe that the results of Equation 8 exhibit minimal variation as $c$ changes. This indicates that the choice of constant $c$ has a limited impact on HeterPos. However, when $c$ equals 0, Equation 8 is no longer affected by $\mathbf{x}_v^k$ and $d$, making it challenging for the model to capture the relative positions of nodes and their neighboring nodes, resulting in a decline in model performance.

**Hyperparameter $h$** To assess the impact of the hyperparameter $h$ on HeterPos fairly, we conducted ablation experiments on $h$. Initially, we set the constants $c$ and $k$ to specific values: $c$ was set to 100, and $k$ to 1. We selected values for $h$ from the set $\{2, 4, 8, ..., 1024, 2048\}$ and simultaneously observed how Equation 8 evolved with changing $h$ and the model's accuracy across eight datasets. The results are depicted in Figure 5 and Figure 6. From these experimental findings, it becomes

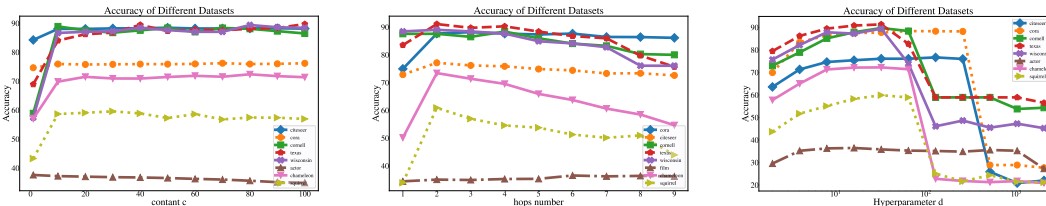

Figure 5: The figure above shows from left to right the curve of accuracy of the model with constant $c$, hyperparameters $k$ and $d$ for the eight datasets.

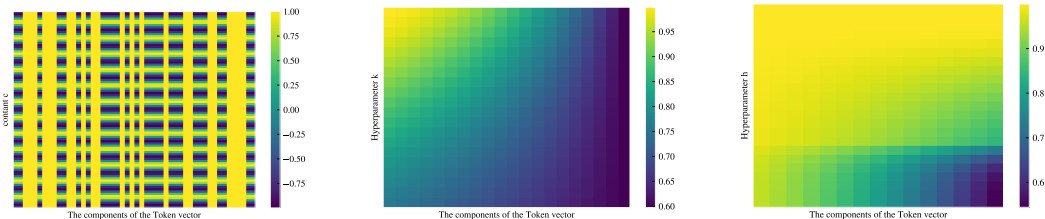

Figure 6: HeterPos with different values of $c$, $k$, $d$. Best viewed in color.

evident that when the model parameter $h$ is less than 8, HeterPos faces challenges in effectively capturing distinctions between nodes and their neighboring nodes. These difficulties arise because node and neighbor information is compressed into small dimensions, making it challenging to represent the differences. However, when $h$ exceeds 8, it allows the model to flexibly represent differences between nodes and neighboring nodes by controlling the size of $h$. However, when $h$ becomes very large (exceeding 1024), it becomes insensitive to changes in the size of $h$, making it challenging to capture subtle distinctions between nodes and neighboring nodes, thereby limiting the model's performance.

**Hyperparameter $k$**   To elucidate the impact of the hyperparameter $k$ on HeterPos, we conducted an ablation experiment. We set the constants and hyperparameters to fixed values: $h$ at 100 and $k$ at 128. Since $k$ signifies the shortest path distance between a node and its neighboring nodes, we visualized how $k$ influences HeterPos as the distance between a node and its neighbors increased. The results are presented in Figure 5 and Figure 6. Upon observing the figure, it becomes evident that our parameter $k$ effectively characterizes the relative positional relationships between nodes and their neighbors. Moreover, as $k$ increases, these distinctions become even more pronounced. However, in practice, the nodes in the $k$-th hop have less influence on the center node due to the fact that the nodes in the $k$-th hop have less influence on the center node as $k$ continues to increase; on the other hand, the number of nodes in the $k$-th strip will grow exponentially as $k$ increases, which leads to a large amount of redundant information participating in the aggregation, thus limiting the performance of the model.

## 4   CONCLUSION

In this paper, we introduce MPformer, a novel model designed to tackle the challenge of extending the existing graph Transformer architecture to heterophilous datasets. This approach constructs a sequence for each node, incorporating its neighbor node features by aggregating nodes at various hops. Additionally, we introduce an innovative and effective positional encoding method that leverages hop number and node representation to embed information, including shortest path distances, similarities between nodes and their neighbors, relative positions, and node heterophily, into the Transformer architecture. We also mathematically illustrate that the positional encoding facilitates a more favorable trade-off between stability and empirical risk minimization, contributing to improved model performance on heterophilous datasets. Our approach surpasses several current state-of-the-art methods in experiments conducted on diverse datasets.

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
