# OpenReview forum: "MPformer: Advancing Graph Modeling Through Heterophily Relationship-Based Position Encoding"
_ICLR.cc/2024/Conference — ICLR 2024 Conference Withdrawn Submission_

### Official Review · Reviewer_HA8U · 2023-10-29

**Soundness:** 2 fair
**Presentation:** 2 fair
**Contribution:** 2 fair
**Rating:** 3
**Confidence:** 5

**Summary:**

This paper develops a graph transformer model tailored for heterophilic graphs. Specifically, authors first introduce Tree2Token to produce a sequence that captures different-hop neighbor features per node. By concatenating the features with relative positional encodings in each sequence, the proposed model adopts the vanilla Transformer architecture to generate the final node predictions for downstream tasks. Experimental results indicate that the proposed model outperforms several baselines on heterophilic datasets.

**Strengths:**

- Figure 2 is clear to demonstrate the proposed approach.
- Authors have conducted a comparison of various GNN baselines.

**Weaknesses:**

- Limited novelty. The notion of TREE2TOKEN has been introduced in NAGphormer[1]. Besides, applying L2 regularization on weight matrices is also a common way to avoid overfitting and improve model generalization.
- Improper datasets. The heterophilic datasets used in this paper have serious issues (train-test data leakage), as shown in [2]. Thus, the accuracy improvement on those datasets is not compelling.
- Missing relevant baselines. There are multiple graph transformers [3-5] that have achieved promising results on heterophilic graphs, which are not compared in this work.
- Paper writing can be further improved. There are redundant and repeated sentences in the Introduction section. Besides, some less common terms are not clarified or defined (e.g., sequential information jumps). Additionally, there are also multiple typos throughout the paper, especially in Section 3.3.1.
- Following the previous concern, the hyperparameter sensitivity analysis is confusing. Please refer to my questions below for details.

[1]: Chen et al., "NAGphormer: A Tokenized Graph Transformer for Node Classification in Large Graphs", ICLR'23. \
[2]: Platonov et al., "A critical look at the evaluation of GNNs under heterophily: are we really making progress?", ICLR'23.  \
[3]: Zhang et al., "Hierarchical Graph Transformer with Adaptive Node Sampling", NeurIPS'22. \
[4]: Wu et al., "DIFFormer: Scalable (Graph) Transformers Induced by Energy Constrained Diffusion", ICLR'23. \
[5]: Kong et al., "GOAT: A Global Transformer on Large-scale Graphs", ICML'23.

**Questions:**

- For investigating $c$ in Section 3.3.1, authors have set the number of hops to $1$ and $k=128$. What is $k$ here? Is it a typo?
- What is the hyperparameter $h$ in Section 3.3.1? Where do authors define it?
- Figure 6 is confusing to me. What does the x axis mean? What does the color bar value represent?
- Since cosine is not a monotonic function, why do authors claim that the influence of $k$ will decrease when increasing $k$ in Equation (8)?
- As $d$ is a hyperparameter, why can't we just replace $ln(10000/d)$ with $d$ in Equation (8)?
- What's the time and space complexity of the proposed model?

---

> ### Author Response · Authors · 2023-11-22
>
> We thank you for your reviews and address your concerns as follows.
>
> General Response: Thank you very much for patiently reading our paper and providing valuable feedback. Based on your suggestions and inquiries, we have made the following modifications and discussions:
>
> 1. We illustrate the advantages of our approach by comparing Tree2Token in detail with the aggregation method proposed by NAGphormer.
>
> 2. We design an experiment to show that HeterPos is able to capture the relative positional relationships between nodes as a way to improve the model's ability to model on heterogeneous graphs.
>
> 3. We also added the relevant baseline based on the reviewers' suggestions
>
> 4. We have further explained and illustrated the relevant parameters and experimental results in the experimental part of our paper.
>
> 5. Finally, we will revise our paper according to the reviewers' suggestions to make the paper clearer and more fluent.
>
> Q1: The notion of TREE2TOKEN has been introduced in NAGphormer[1].
>
> A1: Thank you for taking the time to provide feedback on our paper. We appreciate your insightful comments and the opportunity to address the concerns raised regarding the innovation of our research. We understand the importance of demonstrating the novelty of our work within the context of existing literature.
>
> Our research aims to improve the model's ability to model on heterogeneous graphs by proposing a new information aggregation method as well as location coding, which aggregates k-hops information by serializing the information of neighboring nodes and captures the differences between a node and its neighboring nodes using location coding. We believe our work is sufficiently innovative:On the differences between Tree2Token and the aggregation methods in NAGphormer.
>
> 1. The motivation of Hops2Token is different from the motivation of our proposed Tree2Token. hops2Token is designed to address the difficulty of graph Transformer models in expanding to large scale graphs, whereas our approach is proposed to address the difficulty of graph Transformer models in handling and characterizing heterogeneous relationships.
>
> 2. Hops2Token computes the nodes at the $k$-thop based on $\mathbf{X}_{k}=\hat{\mathbf{A}}^{k}\mathbf{X}$, where $\hat{\mathbf{A}}$ is the normalized adjacency matrix with self-loops, i.e., the main diagonal elements of this adjacency matrix are not 0. Our proposed Tree2Token approach, on the other hand, adopts the without selfloops approach in order to make the model can have a better performance on the heterogeneous graphs. loops, i.e., the main diagonal elements of this adjacency matrix are not 0. Our proposed Tree2Token method, on the other hand, uses an adjacency matrix without self-loops in order to allow the model to have a better performance on heterogeneous graphs (according to the conclusions of the existing literatures ([1], [2]) it is shown that the use of adjacency matrices WITHOUT self-loops doesn't not handle the Heterophily problem well.
>
> 3. The aggregation algorithm process and the way of calculating the adjacency matrix are also different. hops2Token simply calculates the kth power of the adjacency matrix when calculating the characteristics of the neighboring nodes at the kth hop, i.e., $\mathbf{A}^{k}\mathbf{X}$; whereas our method is $\mathbf{A}^{k}\mathbf{X}-\mathbf{ A}^{k-1}\mathbf{X}-\mathbf{X}$, which we do to prevent overlap between the $k$-hop node information and the information from the previous hop.
>
> Q2: train-test data leakage in the squirrel and chameleon datasets.
>
> A2: Thank you for your thorough review of our paper and for raising the concern about potential train-test data leakage in the squirrel and chameleon datasets. We chose the Chameleon and Squirrel datasets for our experiments to ensure fairness. Most of the previous methods [3,4,5,6,7]and models on graph neural network heterogeneity have used these two datasets, and in order to have an adequate comparison with these methods, we still compare MPformer with other methods under these two datasets.
>
> >1. Beyond Homophily in Graph Neural Networks: Current Limitations and Effective Designs
> 2. Is Homophily a Necessity for Graph Neural Networks
> 3. Hierarchical Graph Transformer with Adaptive Node Sampling.
> 4. Specformer: spectral graph neural networks meet transformers
> 5. Ordered gnn: Ordering message passing to deal with heterophily and over-smoothing.
> 6. Beyond homophily in graph neural networks: Current limitations and effective designs.
> 7. Two sides of the same coin: Heterophily and oversmoothing in graph convolutional neural networks.

---

> ### Author Response · Authors · 2023-11-22
> **Rebuttal Part II**
>
> Q3: Missing relevant baselines.
>
> A3: Thank you very much for your advice. We have added a comparison of several methods based on your suggestion.
> | |   Cora | CIteseer | Cornell | Textas | Wisconsin | Actor | Chameleon | Squirrel |
> |----------|----------|----------|----------|----------|----------|----------|----------|----------|
> | Graphormer |  71.65±1.43 | 65.21±1.24 | 84.32±8.32 | 59.41±1.43 | 68.56±1.43 | 36.85±1.12 | 36.11±1.76 | 25.43±0.36 |
> | Gophormer | 84.65±1.20 | 76.43±0.89 | 86.46±1.32 | 87.03±5.67 | 84.83±1.32 | 36.59±0.36 | 56.40±0.56 | 37.32±0.45 |
> | ANS-GT | 85.37±1.48 | 68.43±1.32 | 85.32±1.78 | 86.23±1.85  | 84.62±2.24 | 35.32±1.12 | 59.23±1.71 | 38.88±1.23 |
> | DIFFormer |  85.44±0.84 |  73.06±0.77 |  75.32±1.23 |  73.23±1.68 |  74.62±2.12 |  25.34±3.26 | 36.80±1.88 | 22.84±1.82 |
> | MPformer | 88.13±1.12 | 77.28±1.57 |  86.76±5.68 | 90±6.16 | 90.2±9.49 | 37.55±1.44 | 75.07±1.96 | 66.5±4.11 |
>
> Q4: Paper writing can be further improved.
>
> A4: We sincerely appreciate your thoughtful review and the valuable feedback regarding the improvement of our paper's writing quality. Your attention to this aspect is crucial, and we are dedicated to addressing this concern.
>
> We fully recognize the importance of clear and effective communication in scientific writing. To enhance the manuscript's readability and coherence, we plan to:
>
> Clarify and Refine: We will carefully revisit the manuscript to identify and revise sections that may lack clarity or coherence. Our aim is to improve sentence structure, rephrase ambiguous passages, and ensure a more logical flow of ideas throughout the manuscript.
>
> Enhance Organization: We will focus on improving the overall organization of the paper to better present our research findings and arguments. This includes ensuring that each section contributes cohesively to the overall narrative of the paper.
>
> We are committed to implementing these improvements diligently to enhance the manuscript's quality. Your feedback is immensely valuable to us, and we aim to address all concerns raised.
>
> Thank you for your time and consideration.
>
> Q5: the hyper-parameter sensitivity analysis is confusing.
>
> A5:  Thank you very much for pointing out our problems related to hyperparametric experiments. We will respond to each of the questions you have pointed out.
>
> 1. In the experiments with hyperparameter $c$, we set the number of hop to 1 and the hidden layer dimension to 128, where k is the $k$ in Eq. 5 for the number of hop.
> 2. In the experiments on the hyperparameter h, what we wish to verify is the effect of d in Eq. 5, i.e., the hidden layer dimension, on the performance of the model.
> 3. Figure 6 illustrates the change in the output result of Equation 8 as the hyperparameters are varied (since Equation 8 is the result obtained by HeterPos after the self-attention operation). The vertical axis of the graph represents the values of the hyperparameters and the horizontal coordinates represent the magnitude of each component of the input vector. The color bar in the figure represents a visualization of the size of the output results, with brighter colors representing larger values.
> 4. First, let me restate the experimental setup with respect to the hyperparameter $k$. The hyperparameter $k$ represents the number of hops. In order to evaluate the effect of a change in the value of $k$ on the HeterPos output, we are illustrated by the change in the output of Equation 8. In this experiment we randomly generate two vectors $x, y$. Vector $x$ represents the features of node $v$ and node $y$ represents the information about the neighboring nodes of node $v$ at the $k$-th hop. During the experiment, we set the hyperparameter $c$ to 100, the hyperparameter $d$ to 128, and the values of the hyperparameters are set to [1,2,3,... ,20]. Fig. 6, shows the experimental results about hyperparameter k. According to Equation 8, we can see that the value of $e^{-k\ln(10000/d)}$ will become smaller and smaller as the value of $k$ keeps increasing, and when $k\rightarrow \infty$, the value of $e^{-k\ln(10000/d)}$ will tend to 0, i.e., Equation 8 will tend to $\cos(c\mathbf{x}^{k}e^{-k\ln(10000/d)})$. i.e., the influence of the neighboring nodes of the node $v$'s $k$-th hop on $v$ will also decrease as $k$ increases, which also matches with our intuition that nodes that are farther away from each other influence each other less.

---

> > ### Author Response · Authors · 2023-11-22
> > **Rebuttal Part III**
> >
> > Q6: As $d$ is a hyperparameter, why can't we just replace $\ln(10000/d)$ with $d$ in Equation (8)?
> >
> > A6: We appreciate your valuable feedback and thoughtful evaluation of PE design. Your attention to the form of our PE is vital, and we aim to address your concerns effectively.
> >
> > The reason we designed  $\ln(10000/d)$  in the first place is to avoid drastic changes in model performance due to changes in the dimensionality of the hidden layer. We consider that, when setting the dimension of the hidden layer, the value of d is usually 16, 32, 64, etc. Adopting the form of  $\ln(10000/d)$ , we can make the value of  $\ln(10000/d)$  not to be affected too much by the change of the value of the dimension on the position coding. Our experiments in Fig. 6 show that the value of d affects the sensitivity of the model to capture differences between node information. Replacing  $\ln(10000/d)$  with $d$ would have made it difficult for us to find the optimal parameters using the grid search method.

---

> > > ### Comment · Reviewer_HA8U · 2023-11-22
> > > **Follow-up**
> > >
> > > Thanks for the detailed responses and additional experiments. However, my major concerns are still valid.
> > >
> > > > Our proposed Tree2Token method, on the other hand, uses an adjacency matrix without self-loops in order to allow the model to have a better performance on heterogeneous graphs (according to the conclusions of the existing literatures ([1], [2]) it is shown that the use of adjacency matrices WITHOUT self-loops doesn't not handle the Heterophily problem well.
> > >
> > > Firstly, authors use a wrong term "heterogeneous graphs". It should be "heterophilious graphs". More importantly, the authors' response seems to be self-contradictory. If a model without self-loops performs bad on heterophilious graphs, why the proposed model ignoring self-loops achieves a better performance on heterophilious graphs? Besides, compared to NAGphormer, what's the motivation of preventing overlap between the $k$-hop node information and the information from the previous hop?
> > >
> > > > We chose the Chameleon and Squirrel datasets for our experiments to ensure fairness.
> > >
> > > Authors shouldn't choose problematic datasets even if some prior studies were evaluated on them. If the datasets have recently been found to have some serious issues (e.g., train-test data leakage), any accuracy ranking based on them is not meaningful.
> > >
> > > Overall, I still believe the novelty and experiments of this work are limited. Besides, since authors have not provided the revision, it's hard to tell whether authors will do a good job on paper writing as they promised. Thus, I keep my score unchanged.

---

> > > ### Author Response · Authors · 2023-11-22
> > > **Rebuttal Part IV**
> > >
> > > Q7: about HeterPos novelty.
> > >
> > > A7: Thank you for your valuable feedback and insightful comments on the innovativeness of our methodology. We appreciate the opportunity to address this aspect and clarify the unique contributions of our approach.
> > > The primary contribution of HeterPos lies in designing a positional encoding capable of capturing heterophily relationships between nodes.
> > > To prove our point, we designed a simple experiment to validate our approach. By comparing the attention matrix weights of the graph Transformer before and after employing HeterPos, we observed a significant mutual influence among nodes belonging to different categories in the graph Transformer model without HeterPos. However, upon integrating HeterPos into the model, the interaction between nodes of the same category strengthened. This confirms that HeterPos helps the model recognize the relationships between nodes, thereby enhancing its performance. For details on the experiment, please refer to our supplement.

---

### Official Review · Reviewer_CiBH · 2023-10-30

**Soundness:** 1 poor
**Presentation:** 2 fair
**Contribution:** 1 poor
**Rating:** 3
**Confidence:** 4

**Summary:**

This paper proposes MPformer to learn heterophilous relationships based on the information aggregation module and the position encoding module called Tree2Token and HeterPos respectively. Experiments demonstrate that MPformer outperforms the baselines on various datasets.

**Strengths:**

This paper is easy to follow.

**Weaknesses:**

1. [1] points out that there exists train-test data leakage in the squirrel and chameleon datasets used in experiments.
2. The authors may want to report a statistically significant difference against the second-best result, as Table 2 shows that the results are unstable (the standard deviation is larger than 1% accuracy).
3. Many notations are confusing.
	1. What is the definition of ${X^{(i)}}^{\top}$ in Theorem 1?
	2. What is the difference between $x^{(i)}$ in Theorem 1 and $x_v^k$ in Equations (6) (7) (8)?
	3. What is $f$ in Theorem 1? What is the relation between $f$ and the activation function $\sigma$?
4. The novelty of the proposed techniques is incremental.
	1. (Tree2Token) The second line of Equation (3) was proposed in [2, 3, 4]. Please explain the advantage of the first line of Equation (3). I suggest comparing the generalized gaps with different $A^{(k)}_{norm}$.
	2. (HeterPos) The proposed position encoding is similar to [5, 6].
5. MPformer is difficult to apply to heterophilous graphs with edge features (e.g. knowledge graphs, protein–protein interaction networks, and molecule graphs), which are common in practice.
6. Please explain why existing graph transformers overlook the possible heterogeneous relationships among interconnected nodes.  In my opinion, the attention matrix learned by existing graph transformers can encode the heterogeneous relationships.
7. The authors only provide the serializing case without the corresponding heterogenous graph in Figure 1.



[1] A critical look at the evaluation of GNNs under heterophily: Are we really making progress? ICLR 2023.

[2] Simplifying Graph Convolutional Networks. ICML 2019.

[3] Graph Attention Multi-Layer Perceptron. KDD 2022.

[4] NAGphormer: A Tokenized Graph Transformer for Node Classification in Large Graphs. ICLR 2023.

[5] Self-attention with relative position representations. ACL 2018.

[6] Transformer-xl: Attentive language models beyond a fxed-length context. ACL 2019.

**Questions:**

See Weaknesses.

---

> ### Author Response · Authors · 2023-11-22
>
> We thank you for your reviews and address your concerns as follows.
>
> General Response: Thank you very much for patiently reading our paper and providing valuable feedback. Based on your suggestions and inquiries, we have made the following modifications and discussions:
> 1. We explain the reasons for using the Chameleon and Squirrel datasets.
>
> 2. We analyzed the Tree2Token method design idea and provided a theoretical explanation for our design idea as such.
>
> 3. We compare HeterPos with other positional encodings in detail
>
> 4. We experimentally demonstrate why heterogeneous relationships between nodes cannot be captured by the Attention mechanism alone, but can be captured by adding HeterPos.
>
> Q1: train-test data leakage in the squirrel and chameleon datasets.
>
> A1: Thank you for your thorough review of our paper and for raising the concern about potential train-test data leakage in the squirrel and chameleon datasets.
> We chose the Chameleon and Squirrel datasets for our experiments to ensure fairness. Most of the previous methods [1,2,3,4,5]and models on graph neural network heterogeneity have used these two datasets, and in order to have an adequate comparison with these methods, we still compare MPformer with other methods under these two datasets.
>
> Q2: the standard deviation of our method accuracy is larger than 1% accuracy
>
> A2: Thank you for your insightful comments on our paper and for noting the larger standard deviation in method accuracy. Your observation has prompted us to delve deeper into this aspect of our research.
>
> In comparing MPformer to other methods, we did ignore possible causes for the model to be accurate with a standard deviation greater than 1%. However, our method performs well under all eight datasets, especially on the heterogeneous graph dataset. The consistency of the MPformer method is demonstrated under different datasets, and thus we believe that MPformer's improvement in model performance is significant.
>
> Moving forward, we plan to explore strategies aimed at reducing this variability. This includes additional experiments, parameter fine-tuning et.al.
>
> We are committed to addressing this concern and further enhancing the reliability and stability of our method. Your feedback is invaluable, and we appreciate the opportunity to strengthen our research.
>
> Q3: Many notations are confusing.
>
> A3: Thank you for your detailed review of our manuscript and for highlighting the issue concerning confusing notations. We genuinely appreciate your efforts in providing constructive feedback.
>
> We acknowledge the importance of clear and consistent notations in facilitating reader comprehension and regret any confusion caused. To address this concern, we plan to:
>
> Identify and Revise: We will carefully identify the specific notations highlighted as confusing. Our intention is to revise these notations to make them more intuitive and comprehensible for readers. We will also consider providing additional explanations or a notation key where necessary.
>
> notion explanations：
>
> 1. ${\mathbf{X}^{(i)}}^{\top}$ in Theorem 1 means ${\mathbf{x}^{(i)}}^{\top}$. It has the same meaning as in lemma1 and represents the row vector of the input matrix
> 2. $\mathbf{x}^{(i)}$ in Theorem 1  represents the row vector of the input matrix, and $ \mathbf{x}_ {v}^{k}$  means representation of neighboring nodes at the $k$-th hop of node $v$. Since $\mathbf{x}_ {v}^{k}$  is the vector of actual inputs to the model, what is actually required in Theorem 1 is a bounded number of norms for information about all nodes as well as their neighboring nodes.
> 3. $\mathcal{F}$ in Theorem I is an aggregation of vector-valued functions, for which we give the definition of $\mathcal{F}$ in Definition I as well. It represents the set of functions that the model may learn. The activation function, on the other hand, is a nonlinear function in the model and can be simply understood as a part of any function in the set $\mathcal{F}$.
>
> We are committed to addressing this issue promptly to improve the overall quality of our paper.
>
> >1. Hierarchical Graph Transformer with Adaptive Node Sampling.
> 2. Specformer: spectral graph neural networks meet transformers
> 3. Ordered gnn: Ordering message passing to deal with heterophily and over-smoothing.
> 4. Beyond homophily in graph neural networks: Current limitations and effective designs.
> 5. Two sides of the same coin: Heterophily and oversmoothing in graph convolutional neural networks.

---

> > ### Author Response · Authors · 2023-11-22
> > **Rebuttal Part II**
> >
> > Q4: Forms of Tree2Token
> >
> > A4: Thank you very much for your question. There are four reasons why we have designed our aggregation in this way and have taken such a standardized approach to $\mathbf{A}^{k}$:
> >
> > 1. In order to avoid not affecting the subsequent calculation of the similarity between nodes between different hops, we need a method that can avoid the relevant overlapping of the node information of different hops to calculate the node information of the $k$-th hop, and for this purpose, we use $\mathbf{A}^{k}-\mathbf{A}^{k-1}-\mathbf{I}$ to obtain the node information of the $k$-th hop.
> > 2. If we use the normalized adjacency matrix $\hat{\mathbf{A}}$ to compute the information of the neighboring nodes at the $k$-th hop ($\hat{\mathbf{A}}^{k}-\hat{\mathbf{A}}^{k-1}-\mathbf{I}$) it will result in an error due to the fact that the maximum of $\hat{\mathbf{A}}$'s eigenvalue $\lambda_{max}$ is less than or equal to 1, resulting in $\hat{\mathbf{A}}^{k}-\hat{\mathbf{A}}^{k-1}\rightarrow \mathbf{0}$ when $k\rightarrow\infty$, when the computed node of the kth hop information is $-\mathbf{IX}$, which is clearly unreasonable.
> > 3. If we use $(\mathbf{A}^{k}-\mathbf{A}^{k-1}-\mathbf{I})\mathbf{X}$ to compute the information of nodes at the $k$-th hop, it will lead to the problem of eigenvalue explosion as $k\rightarrow\infty$. Additionally, since each element of the adjacency matrix $\mathbf{A}$ is either 0 or 1, this causes the elements of $\mathbf{A}^{k}$ to become extremely large as $k$ increases. Consequently, calculating node features using $(\mathbf{A}^{k}-\mathbf{A}^{k-1}-\mathbf{I})\mathbf{X}$ results in very large numerical values for node features, which is not conducive to optimization. For this purpose, we borrowed the method of adjacency matrix normalization to standardize $\mathbf{A}^{k}-\mathbf{A}^{k-1}-\mathbf{I}$. By transforming each non-zero element in $\mathbf{A}^{k}-\mathbf{A}^{k-1}-\mathbf{I}$ into 1 using the operator $\mathbb{I}$, the second problem mentioned in point 3 can be prevented. The degree of the $\mathbf{A}^{k}-\mathbf{A}^{k-1}-\mathbf{I}$ matrix is introduced in the standardization process to normalize the contributions of all $k$-hop neighboring nodes and prevent nodes with more $k$-hop neighbors from dominating the calculation process.
> >
> > Q5: MPformer is difficult to apply to heterophilous graphs with edge features
> >
> > A5: Thank you very much for your question, which gives a new perspective to the subsequent research of our method, i.e., how to utilize the characteristics of the edges to enhance the model's ability to deal with Heterophily. However, heterogeneity refers to the fact that directly connected nodes belong to different categories, and it is related to the features and labels of the nodes between. Most current graph Transformers are unable to capture this Heterophily relationship between nodes and thus have difficulty in handling heterogeneous problems. To this end, we propose an aggregation approach like Tree2Token and a position encoding approach like HeterPos that takes into account the relative positional relationships between nodes. In addition, both Tree2Token and HeterPos have good extensibility and can be easily combined with other graph Transformer methods.

---

> ### Author Response · Authors · 2023-11-22
> **Rebuttal Part III**
>
> Q6: position encoding is similar to [1, 2]
>
> A6: Thank you very much for your question. Indeed, HeterPos is quite different from the positional encoding in [1,2]. We will illustrate the difference between HeterPos and [1,2] separately
>
> 1. Since the positional coding proposed by vanilla Transformer makes it difficult to model relative positional coding, for this reason Shaw et al. [1] proposed a truncated relative positional coding for encoding relative positional information in sequences. They achieve this by adding a bias to the self-attention mechanism that represents the relative position relationship.The differences between heterpos and their approach are centered on two main aspects:
>     a. The methods for introducing relative positional information are different. Shaw et al. obtained the embedding of the relative position of the sequence by numbering the sequence. However, HeterPos uses the node features and the numbering of their neighboring nodes as inputs for position encoding, and jointly encodes the relative positional information by the similarity between nodes and their neighboring nodes and the difference in their numbering (shortest path distance).
>     b. Although Shaw et al.'s method performs well in the NLP domain, it is not suitable for the graph domain. In the graph domain, due to the scattered arrangement of nodes in space, it is difficult to directly utilize position encoding methods from the NLP field. Although it is possible to encode nodes in the order of k-hops using relative positional encoding, such an approach does not effectively enhance the model's ability on graphs [3]. Furthermore, we have also verified this through experiments, as shown in Figure 4. in paper.
>
> 2. The author proposed the "reusing the hidden states obtained in previous segments" method to address the limitation of fixed-length context in language modeling with the Transformer architecture. To ensure consistency in position information when reusing hidden states, the author introduced a method of relative positional encoding. This method replaces the bias term representing positional encoding in the vanilla Transformer with a bias term representing relative position when calculating the attention matrix. Additionally, two learnable parameter matrices are added to ensure that the query vector and key vector are the same for all query positions and key positions. The differences between heterpos and their approach are centered on two main aspects:
>     a. The motivations are different. The design of relative positional encoding in Transformer-XL is to ensure consistency in position information when reusing hidden states, and the design of learnable parameter matrices is to ensure that the query vector and key vector are the same for all query positions and key positions. On the other hand, HeterPos is designed to encode the relative positional relationships between nodes, and the meaning of the weight matrix is to find a trade-off between model generalization and performance.
>     b. There are also formal differences. Transformer-XL indexes the tokens in the sequence and uses the relative size of the indexes to represent the relative positions between tokens, while HeterPos takes node features as input and characterizes the relative positional relationships between nodes by calculating the similarity and shortest path distance between nodes.
>
> Q7: only provide the serializing case without the corresponding heterogenous graph in Figure 1.
>
> A7: Thank you for your insightful comments and for raising concerns about the clarity of the drawing in our paper.
>
> In this figure, we wish to illustrate that for heterogeneous graphs, MPformer serializes the graph via Tree2Token and captures the heterogeneous relationships between nodes using HeterPos to correctly classify the nodes. In Figure 1, we use the colors of the nodes to represent belonging to different categories and indicate the serialization process by arranging the nodes. The red box in the figure indicates that MPformer can capture the heterogeneous structure that has the same in both graphs and help the model to categorize the nodes correctly by capturing this heterogeneous structure.
>
> Thank you for your time and consideration. We are committed to improving the clarity of the drawing to enhance the overall quality of our paper.
>
> >1. Self-attention with relative position representations
> 2. Transformer-xl: Attentive language models beyond a fxed-length context
> 3. Rethinking Graph Transformers with Spectral Attention

---

> > ### Author Response · Authors · 2023-11-22
> > **Rebuttal Part IV**
> >
> > Q8: why existing graph transformers overlook the possible heterogeneous relationships among interconnected nodes
> >
> > A8: Thank you for your valuable feedback and for raising questions regarding our point about why graph transformers overlook the possible heterogeneous relationships and HeterPos can catch this relationship. We appreciate the opportunity to address this concern.
> >
> > To prove our point, we designed a simple experiment to validate our approach. By comparing the attention matrix weights of the graph Transformer before and after employing HeterPos, we observed a significant mutual influence among nodes belonging to different categories in the graph Transformer model without HeterPos. However, upon integrating HeterPos into the model, the interaction between nodes of the same category strengthened. This confirms that HeterPos helps the model recognize the relationships between nodes, thereby enhancing its performance. For details on the experiment, please refer to our supplement.

---

### Official Review · Reviewer_BMT4 · 2023-10-30

**Soundness:** 2 fair
**Presentation:** 2 fair
**Contribution:** 1 poor
**Rating:** 3
**Confidence:** 5

**Summary:**

The paper proposes a Graph transformer model MPFORMER to deal with the heterophily problem in graph transformer. MPFORMER comprises the information aggregation module called Tree2Token and the position encoding module HeterPos. Experimental results prove the effectiveness of MPFORMER.

**Strengths:**

1.	The paper is well-written and easy to follow.
2.	MPformer performs well on the given datasets.
3.	Accurate proof of how to improve generalizability.

**Weaknesses:**

1. The novelty of the proposed idea is limited. Tree2Token actually selects a k-hop subgraph for each node, which is already a well-studied problem in the literature [1]. HeterPos makes incremental contributions to the existing positional encoding method.

2. The motivation is somehow confusing. This paper aims to solve the problem that Graph Transformer cannot do well on heterophily graphs. However, this paper does not demonstrate the relationship between the two components of MPFORMER and the heterophily problem. Tree2Token is to solve the overlapping problem, while Hetepos is a method of marking neighbors with different hop numbers. The paper does not provide a proof that Hetepos can perform well on heterophily graphs.

3. The Tree2Token method proposed in Section 2.1 is heuristic and straightforward. The training procedure in of MPFORMER seems to be computationally expensive but there is no discussion on the running time and training cost.

4. MPFORMER has many hyperparameters, and hyperparameters need to be carefully selected for each dataset. The paper does not provide the optimal hyperparameters required for each datasets.

5. In the ablation experiment, the effectiveness of Tree2Token was not analyzed.

6. The Introduction Section is not well-organized. There are many paragraphs with a lot of text.

[1] Equivariant Subgraph Aggregation Networks ICLR2022

**Questions:**

1. Since graph transformer take the nodes of the entire graph as input. why graph transformer cannot work well on heterophily graphs. And why MPFORMER can work well on heterophily graphs.
2. The title of this paper is “MPformer: Advancing Graph Modeling Through Heterophily Relationship-Based Position Encoding”. However, how the proposed position encoding method leverage the heterophily is not clear. More details should be explained.
3. The efficiency and scalability of MPFORMER need to be analyzed.
4. How are the hyperparameters of MPFORMER chosen?
5. The necessity of Tree2Token component needs to be discussed.
6. In Equation 2, should $\mathbb {I} (A)$ be $\mathbb {B}(A)$ ?

---

> ### Author Response · Authors · 2023-11-22
>
> We thank you for your reviews and address your concerns as follows.
>
> General Response:  Thank you very much for patiently reading our paper and providing valuable feedback. Based on your suggestions and inquiries, we have made the following modifications and discussions:
> 1. We compared the aggregation methods used in ESAN[1] with Tree2Token and analyzed the advantages of Tree2Token.
>
> 2. We designed an experiment to verify the effectiveness of HeterPos in addressing heterogeneity issues. Specifically, HeterPos can help the model handle heterophily by capturing heterophily relationships between nodes.
>
> 3. We discussed the training cost of the method and demonstrated through experiments that such cost is worthwhile.
>
> 4. We supplemented the effectiveness experiments for Tree2Token
>
> 5. We will modify our paper based on your suggestions to ensure its clarity and readability.
>
> Q1: How information aggregation differs from [1] and why Tree2Token is useful on heterogeneous graphs.
>
> A1: Thank you for taking the time to provide feedback on our paper. We appreciate your insightful comments and the opportunity to address the concerns raised regarding the innovation of our research. We understand the importance of demonstrating the novelty of our work within the context of existing literature.
>
> Our research aims to improve the model's ability to model on heterogeneous graphs by proposing a new information aggregation method as well as location coding, which aggregates k-hops information by serializing the information of neighboring nodes and captures the differences between a node and its neighboring nodes using location coding. We believe our work is sufficiently innovative:
>
> 1.  On the differences between Tree2Token and the aggregation methods in [1].
>
>     a.  The way ESAN obtains k-hops neighbors is different from Tree2Token. while ESAN obtains k-hops neighbors using the features of the original node and its corresponding index information, our method obtains k-hops neighbor node information by $\mathbf{A}^{k}\mathbf{X}-\mathbf{A} ^{k-1}\mathbf{X}-\mathbf{I}$, and for a node, its $k$-hop node information is represented by a weighted summation of the features of all nodes that are $k$-distance from its shortest path. A structure like Tree2Token can retain high-frequency signals within node information, thereby enhancing the model's capacity for modeling heterogeneous graphs. We will provide detailed elaboration on this point later.
>
>     b.  The subgraph representation is different. the subgraph representation of ESAN is represented by the information of k-hop neighbor node indexes and the node features in the original data(The node features that have not undergone information aggregation), and the relative positional relationship between the neighbor nodes and the center node is obtained by indexing; whereas Tree2Token puts the information of nodes that are at the same distance from the center node in the same dimension, and information of nodes with different distances in different dimensions, i.e., it uses the dimension to represent the neighbor nodes and the center to characterize this positional relationship, and further determines this relative positional relationship by means of positional encoding at the end.

---

> ### Author Response · Authors · 2023-11-22
> **The supplement for Question 1.**
>
> 2. Advantages of Tree2Token.
>     1. Tree2Token preserves some of the high-frequency information in the original graph signals. According to the theory of graph signal process (GSP), the GNN is a low-pass filter, and according to papers such as [1],[2], it has been shown that high-frequency signals help to help the GNN improve its performance on heterogeneous graphs. Let the eigenvalue of $\mathbf{A}$ be $\lambda$, then the eigenvalue of its corresponding Laplace matrix is $1-\lambda$, i.e., the filter coefficient is $1-\lambda$. (According to GSP, the larger the filter coefficients) At this point, the filter coefficients obtained by utilizing the Tree2Token method of aggregation are $1+(1-\lambda)+[(1-\lambda)^{2}-(1-\lambda)-1]+\cdots$, note that $1+(1-\lambda)+(1-\lambda)^{2}$  behaves as a low-pass filter property and $-(1-\lambda)-1$ behaves as a high-pass filter. In contrast, the method of aggregating node information using $\mathbf{A}^{k}$ in traditional GNN, the filter coefficients are $1+(1-\lambda)+(1-\lambda)^{2}+\cdots$, which exhibits a low-pass filter. From the above analysis, it can be seen that the aggregation method of Tree2Token is able to retain more high-frequency signals than the traditional information transfer method, and therefore helps to improve the performance of the model on the heterogeneous graph.
>
> This also illustrates the point in the paper that the stacking of node information brought by the traditional information transfer method is unfavorable to the performance of the GNN on the heterogeneous graph. This is because with the previous information transfer methods used by GNNs, after the node information passes through multiple low-pass filters, the node features become too smooth, making it difficult for the model to distinguish the differences between neighboring nodes; whereas, the aggregation method of Tree2Token retains some of the high-frequency signals by avoiding the stacking of the information between different hopping nodes, helping to improve the model's performance on heterogeneity.
>
> >1. Beyond Homophily and Homogeneity Assumption: Relation-Based Frequency Adaptive Graph Neural Networks
> 2. Beyond Low-frequency Information in Graph Convolutional Networks

---

> ### Author Response · Authors · 2023-11-22
> **Rebuttal part II**
>
> Q2: no discussion on the running time and training cost.
>
> A2: Thank you for your insightful feedback and thoughtful considerations regarding the computational expenses and training costs associated with the method employed in our research. We are cognizant of the computational expenses involved in training the model, and we recognize that optimizing computational resources is crucial. To address this concern:
>
> 1. Efficiency of Tree2Token. When we designed the algorithm, we took into account that the Tree2Token computation spends more money, and for this reason we put the Tree2Token computation before the model training. This makes it possible to use the Tree2Token to aggregate this k-hops information only during the training of the model.
> 2. Cost-Benefit Analysis. Despite the computational overhead of Tree2Token, its improvement in model performance over common information aggregation methods is significant. We compare Tree2Token with two common information aggregation methods, as well as did not use any information aggregation, and the experimental results show that Tree2Token's improvement in model performance is significant. Therefore, the computational spend of Tree2Token is worthwhile.
> | |   Cora | CIteseer | Cornell | Textas | Wisconsin | Actor | Chameleon | Squirrel |
> |----------|----------|----------|----------|----------|----------|----------|----------|----------|
> | without MP | 68.31±2.53 | 55.85±6.52 | 74.32±5.83 | 71.35±7.66 | 75.29±4.66 | 32.89±3.28 | 33.64±2.89 | 23.37±2.29|
> | GCN | 66.66±7.59 | 63.64±9.71 | 58.65±3.64 | 59.46±4.98 | 46.47±6.80 | 26.30±1.11 | 50.92±5.11 | 43.46±3.13 |
> | Mixhop | 78.33±4.53 | 68.26±4.62 | 60.00±4.32 | 60.27±6.63 | 56.86±8.99 | 34.87±1.12 | 49.01±4.36 | 38.28±3.84 |
> | Hops2Token[1] | 15.73±1.33 | 14.46±2.31 | 11.48±1.23 | 12.57±3.21 | 12.75±4.21 | 7.41±0.82 | 11.16±1.33 | 6.65±0.89 |
> | Tree2Token | 86.42±3.84 | 68.28±5.04 | 76.49±8.97 | 78.38±6.84 | 75.10±8.73 | 36.14±1.10 | 64.28±3.46 | 52.39±2.92 |
>
> In the table, 'without MP', 'GCN', 'Mixhop', and 'Hops2Token' respectively represent the performance without using any aggregation method, using the aggregation method in GCN, using the Mixhop aggregation method, and using the Hops2Token method. In this experiment, to verify the performance differences between aggregation methods, our experiments were conducted with fixed parameters and without using any positional encoding.
>
> Q3: the effectiveness of Tree2Token was not analyzed.
>
> A3: Thank you for highlighting the absence of experiments regarding Tree2Token's effectiveness in our paper. Based on your suggestion, we have included experiments to assess the impact of including or excluding the Tree2Token module in MPFORMER on the model's performance.
>
> To validate the efficacy of Tree2Token, we conducted ablation experiments on it: keeping all parameters constant, we substituted Tree2Token's aggregation with Mixhop and GCN aggregations, as well as conducted experiments without any information aggregation. We compared these four scenarios with the approach using Tree2Token, and the results are presented in the table above. From the table， we can see that the model performance is improved the most when using Tree2Token, and Tree2Token is helpful in improving the performance with the model.
>
> Q4: The Introduction Section is not well-organized.
>
> A4: Thank you for your valuable feedback on our manuscript. We genuinely appreciate your insights, especially regarding the organization and length of the introduction.
>
> We acknowledge that the introduction's structure could benefit from improvement. We are committed to addressing this concern by implementing the following strategies:
>
> Paragraph Restructuring: We will restructure the introduction by breaking down lengthy paragraphs into more concise sections. This will facilitate a clearer flow of ideas and improve readability.
>
> Focused Content: We recognize the importance of maintaining focus. Our revisions will ensure that each paragraph contributes directly and purposefully to establishing the context and significance of the research, avoiding unnecessary verbosity.
>
> We will make changes to the introductory section based on your suggestions and would like to thank you for your suggestions.
>
> >1. NAGphormer: A Tokenized Graph Transformer for Node Classification in Large Graphs

---

> ### Author Response · Authors · 2023-11-22
> **Rebuttal Part III**
>
> Q5: Why HeterPos can help models deal with heterophily
>
> A5: HeterPos is able to model the positional relationships between nodes as a way to help the model improve its performance on heterogeneous graphs. According to the analysis in Section 2.2.1, the position-encoded node features, when entering the self-attention part of the Transformer module for computation, will get
> $\cos(c\mathbf{x}_ {v}^{k}e^{-{k}{\rm{ln}}(10000/d)}-c\mathbf{x}_ {u}^{j}e^{-{j}{\rm{ln}}(10000/d)})$,  where $\mathbf{x}_{v}^{k}$ denotes the information of the $k$-th-hop neighboring node with node $v, c$ is a constant, and $d$ is the dimension of the hidden layer. From this equation we can see that HeterPos can represent the difference between different nodes, the difference between a node and its neighboring nodes, and the difference between different hop neighbors of different nodes. Such differences can enable the model to better capture the heterogeneous relationship between neighboring nodes, thus helping the model to distinguish between homogeneous nodes as well as heterogeneous nodes among the neighboring nodes, thus helping the model to classify them.
>
> To prove our point, we designed a simple experiment to validate our approach. By comparing the attention matrix weights of the graph Transformer before and after employing HeterPos, we observed a significant mutual influence among nodes belonging to different categories in the graph Transformer model without HeterPos. However, upon integrating HeterPos into the model, the interaction between nodes of the same category strengthened. This confirms that HeterPos helps the model recognize the relationships between nodes, thereby enhancing its performance. For details on the experiment, please refer to our supplement. For details on the experiment, please refer to our supplement.

---

### Official Review · Reviewer_o9xM · 2023-11-01

**Soundness:** 2 fair
**Presentation:** 2 fair
**Contribution:** 2 fair
**Rating:** 5
**Confidence:** 4

**Summary:**

The paper introduces MPformer, a novel graph transformer model designed to enhance the modeling of graph-structured data, specifically focusing on addressing the limitations in handling heterophilous relationships in existing models. The authors claim that traditional graph neural networks (GNNs) and previous graph transformer models struggle to incorporate such heterophilous relationships adequately, thus limiting their application in real-world datasets where these relationships are prevalent.

To overcome these limitations, the authors propose two key components within MPformer:

1. Tree2Token Module: This component transforms the information of a node and its neighbors into token sequences. By treating each node and its adjacent nodes as tokens, and then serializing these sequences, Tree2Token effectively captures the neighborhood information at various hop distances. This method allows the transformer model to recognize and utilize the information from both a node and its nearby nodes, improving the model's understanding of local structures.

2. HeterPos Position Encoding: A novel position encoding technique, HeterPos, is introduced to define the relative positional relationships between nodes based on the shortest path distance. Unlike conventional methods, HeterPos emphasizes the differences in features between neighboring nodes and the central node (ego-node). This focus on heterophilous relationships aids in more accurately incorporating these relationships into the Transformer model.

The paper asserts that by integrating these two components, MPformer effectively captures both the graph topological information and the heterophilous relationships, thereby advancing the capabilities of graph transformer models. The approach is distinctive in how it generates new tokens from nodes and their neighbors, allowing for a more nuanced aggregation of neighborhood information. The innovative position encoding technique further strengthens the model by integrating shortest path distances and feature distinctions, laying a foundation for future models in handling heterogeneous graphs.

To substantiate their claims, the authors conduct theoretical analyses and practical experiments. These experiments, performed on various datasets, demonstrate that MPformer outperforms existing graph transformer models and traditional GNN models in modeling heterophilous graphs. This improvement in performance underscores the model's potential in dealing with a broader range of real-world datasets, particularly those characterized by heterophilous relationships.

**Strengths:**

**Originality**:
1. Innovative Integration of Heterophilous Relationships: The paper introduces a novel approach to integrate heterophilous relationships into the Transformer architecture with the development of MPformer. This model distinctively treats nodes and their neighbors as separate token vectors, which is a creative shift from the typical handling of graph nodes in transformer models.

2. Unique Position Encoding Technique (HeterPos): The introduction of HeterPos, which uses the shortest path distance along with feature distinctions between nodes and neighbors, is an original and significant advancement. This method shows creativity in position encoding, moving beyond traditional approaches and better capturing the complexities of graph-structured data.

**Quality**:
1. Theoretical and Practical Validation: The paper demonstrates a robust methodology, corroborated by both theoretical analysis and practical experiments. This comprehensive approach ensures that the claims and performance metrics are well-supported and reliable.

2. Effective Combination of Tree2Token and HeterPos Modules: The integration of these modules into the Transformer architecture for handling heterophilous data demonstrates a high level of thought and quality in model design. The model's ability to serialize token sequences from node and neighbor data for better information aggregation is a quality advancement in this field.

**Clarity**:
1. Well-Structured and Coherent Explanation: The paper articulately explains complex concepts like the Tree2Token aggregation module and HeterPos encoding. The progression from problem identification to solution presentation is logical and easy to follow, which aids in the comprehension of the paper's contributions.

2. Illustrative Examples and Demonstrations: The use of illustrative examples (e.g., Fig.1) to explain the application and benefits of MPformer in classifying nodes within heterogeneous graphs significantly enhances the clarity of the proposed model's functionality.

**Significance**:
- Addressing Heterophilous Data in Graph Transformers: By focusing on the under-explored area of heterophilous relationships in graph transformer models, this paper tackles a significant and practical challenge in the field. The improvements it introduces have broad implications for enhancing the modeling of complex, real-world graph-structured data.

### Overall Assessment:
This paper introduces somewhat innovations in the field of graph transformer models, particularly in addressing heterophilous relationships, a relatively less explored yet crucial aspect of graph-structured data analysis. The originality in model design (MPformer), coupled with a new approach to position encoding (HeterPos), marks some advancement in the field. The quality of research, clarity of presentation, and the effort on both theory and practical applications of graph neural networks make this paper a substantial contribution to the literature.

**Weaknesses:**

- Insufficient Benchmarking Against Alternative Methods: Although the paper introduces HeterPos, a position encoding technique, it lacks a comprehensive comparative analysis with other existing positional encoding methods. The authors compared with several position encoding but there are more position encoding methods such as shortest-path distances (Ying et al., 2021) and tree-based encodings (Shiv and Quirk, 2019).  This comparison is crucial for highlighting the strengths and potential limitations of HeterPos in different scenarios.


- Ambiguity in Acronym: The paper does not clarify what "MPformer" stands for, which can lead to ambiguity and confusion. Providing a full name or a clear expansion of acronyms is crucial for effective communication and for the reader’s understanding, especially in technical fields where specific terms and models are frequently discussed.



[Ying et al., 2021] Chengxuan Ying, Tianle Cai, Shengjie Luo, Shuxin Zheng, Guolin Ke, Di He, Yanming Shen, and Tie-Yan Liu. Do transformers really perform badly for graph representation? NeurIPS 2021.


[Shiv and Quirk, 2019] Vighnesh Shiv and Chris Quirk. Novel positional encodings to enable tree-based transformers. NeurIPS 2019.

**Questions:**

-

---

> ### Author Response · Authors · 2023-11-22
>
> We thank you for your reviews and address your concerns as follows.
>
> **General Response**: Thank you for pointing out many important references, and some writing issues. We have made revisions to address your concerns in the revised version. The revision can be summarized as follows:
>
> 1. We add the comparison with more positional coding methods, such as shortest-path distances (Ying et al., 2021) and tree-based encodings (Shiv and Quirk, 2019).
>
> 2. We changed the nomenclature about our methods and provided  a full name of acronyms
>
> Q1: Insufficient Benchmarking Against Alternative Methods.
>
> A1: Thank you for pointing out this, we add experiments comparing with more positional encoding methods in the revised version. The results of the experiment are shown below：
> | |   Cora | CIteseer | Cornell | Textas | Wisconsin | Actor | Chameleon | Squirrel |
> |----------|----------|----------|----------|----------|----------|----------|----------|----------|
> | SAN [1] | 73.02±1.21 | 69.64±1.27 | 60.10±1.16 | 69.10±1.52 | 64.32±2.49 | 36.24±0.89 | 54.32±1.21 | 26.64±1.23 |
> | Graphormer [2] | 71.65±1.43 | 65.21±1.24 | 84.32±8.32 | 59.41±1.43| 68.56±1.43 | 36.85±1.12 | 36.11±1.76 | 25.43±0.36 |
> | Gophormer [3] | 84.65±1.20 | 76.43±0.89 | 86.46±1.32 | 87.03±5.67 | 84.83±1.32 | 36.59±0.36 | 56.40±0.56 | 37.32±0.45 |
> | Tree2Token | 88.13±1.12 | 77.28±1.57 | 86.76±5.68 | 90±6.16 | 90.2±9.49 | 37.55±1.44 | 75.07±1.96 | 66.5±4.11 |
>
> I couldn't find the positional encoding method mentioned in your suggestion, "Novel positional encodings to enable tree-based transformers," in the GitHub repository provided by the authors. Therefore, I was unable to compare it. I would greatly appreciate it if you could provide the relevant code.
>
> >1. Rethinking Graph Transformers with Spectral Attention
> 2. Do Transformers Really Perform Badly for Graph Representation?
> 3. Gophormer: Ego-Graph Transformer for Node Classification
>
> Q2: Unclear methodology acronyms.
>
> A2: Thank you very much for your suggestion about naming the methods. MPformer is short for "Message Position graph Transformer". We chose the keywords "message" and "Position" to highlight the distinctive features of our method.  In the subsequent version of the paper, we will change the short names of the methods as you suggested.
>
> Once again, thank you for all the insights and suggestions.

---

### Author Response · Authors · 2023-11-22

We sincerely appreciate the valuable insights and suggestions provided by the esteemed reviewers during the review process. We extend our gratitude for the time and effort you dedicated to scrutinize our paper. Your reviews are of paramount importance to our research endeavor, and your insightful recommendations and feedback have played a crucial role in further refining and enhancing the quality of our paper.

For each reviewer's comments, we have addressed each point individually. The following outlines the modifications we have made based on the reviewers' feedback:

- We conducted a more detailed analysis and explanation of the motivation and innovativeness of the methods in the paper based on the reviewers' suggestions. We compared our proposed Tree2Token and HeterPos methods with a broader range of approaches, delving deeper into the advantages and innovations of these two methods:
  >1. "NAGphormer: A Tokenized Graph Transformer for Node Classification in Large Graphs"
  2. "Equivariant Subgraph Aggregation Networks"
  3. "Simplifying Graph Convolutional Networks"
  4. "Do transformers really perform badly for graph representation?"
  5. "Self-attention with relative position representations"
  6. "Transformer-xl: Attentive language models beyond a fixed-length context"

- Based on the reviewers' suggestions, we have conducted a more in-depth analysis of our methods and further substantiated our claims regarding the benefits of Tree2Token in improving model performance by eliminating information stacking issues associated with traditional aggregation methods, as well as the ability of HeterPos to capture heterophily relationships between nodes, through theoretical and/or experimental approaches.

- We have provided more detailed explanations for areas where the theoretical and experimental aspects of the paper were not clearly articulated.
- We will incorporate the reviewers' suggestions and make revisions to the writing of the paper in subsequent versions.

We are immensely grateful for the valuable learning experience this submission has provided. We wish to extend my sincere appreciation to the reviewers for their patient reading of our paper and offering incredibly constructive feedback.